# V2A-CoT: A Training-Free Video-to-Audio Method via Expert Chain-of-Thought

## Abstract

The rapid development of multimodal models, such as Sora, has given rise to video generation. To make the generated video more realistic, the sound is expected to be generated along with the video. Existing video-to-audio (V2A) works, however, most rely on training, which incurs tremendous computing costs, making them unsuitable for general scenes. In this paper, we propose V2A-CoT, a zero-shot Chain-of-Thought (CoT) based on multimodal large language models (MLLMs). V2A-CoT understands the video from both macro and micro perspectives and generates corresponding audio. Specifically, V2A-CoT guides the MLLM in capturing object data from the video through a carefully designed expert CoT, predicting its sound attributes from a macro perspective across the entire video. By referencing this data, the MLLM enriches the sound details from multiple perspectives. Then, to predict the dynamic changes in sound, V2A-CoT captures the relative position of the sound source from a micro perspective, ultimately generating the audio using a pre-trained model. Compared to state-of-the-art V2A methods: (i) V2A-CoT is the first training-free V2A method, which enhances video understanding from macro and micro perspectives; (ii) comprehensive evaluations demonstrate that V2A-CoT enhances MLLM's audiovisual understanding ability, improving video understanding accuracy by up to 14%, and the audiovisual consistency of generated audio is nearly doubled; (iii) V2A-CoT excels in finely controlled sound tasks and adapts seamlessly to any MLLM in a plug-and-play manner.

## 1 Introduction

In recent years, with the continuous development of generative models such as generative adversarial networks (GANs) and diffusion models, various advanced generative models (e.g., DiT-based models) have made significant progress in multimodal video generation tasks such as text-to-video and image-to-video generation (Wu et al., 2023; Jiang et al., 2024), with models like Sora leading the way (Liu et al., 2024). Although the videos generated by these models exhibit high coherence, detail fidelity, and overall clarity, most of these videos currently lack or are unable to provide audio that is perfectly synchronized with the video content, which significantly diminishes the immersion and realism of the generated videos.

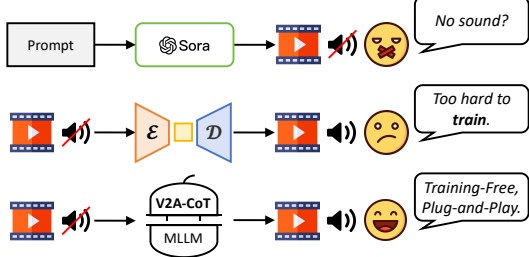

Figure 1: Comparison of audiovisual generation methods: 1) *existing models:* generate video-only content without synchronized audio; 2) *current V2A methods:* require costly training for video-to-audio conversion; 3) *our solution:* enables training-free video dubbing.

To enhance the realism of these generated videos, some mainstream methods have turned to video-to-audio (V2A) methods (Ren et al., 2024; Yang et al., 2024). The core challenge lies in how to understand the video content using only silent video frames and how to generate the corresponding audio from the understanding of the video content (including attributes such as pitch, volume, duration, and timbre for different objects). As shown in Figure 1, previous work has been training-based, such as Stable-V2A (Gramaccioni et al., 2024),

which requires targeted training on specific datasets, such as Greatest Hits (Owens et al., 2016) and AVQA (Yang et al., 2022), to achieve the desired dubbing effect (Zhang et al., 2024; Mo et al., 2024; Lee et al., 2024; Jeong et al., 2025). However, training these models incurs high computational and time costs, making these training-based V2A methods difficult to implement.

Recently, large language models (LLMs) and multimodal large language models (MLLMs), combined with the Chain-of-Thought (CoT) method, have opened up more possibilities for solving multimodal problems at a lower cost (Mondal et al., 2024; Zheng et al., 2023). The goal of CoT is to enhance the ability of large models to handle complex tasks. The core idea is to decompose a complex task into a series of logical steps to ultimately derive a solution, which is a structured problem-solving mechanism (Wei et al., 2022). Current CoT methods are mainly divided into zero-shot (no examples) and few-shot (with examples), with zero-shot being more suitable for meeting diverse needs without manually designing instances, making it ideal for video dubbing.

Inspired by these characteristics, we propose V2A-CoT, a video-to-audio framework based on video understanding via expert CoT. V2A-CoT consists of two steps: video understanding and audio generation. In the video understanding phase, an expert CoT first prompts the MLLM to automatically capture all object data in the video scene from a macro perspective, avoiding forgetting. Then, the obtained object list and specific questions are input into the MLLM in JSON format to obtain detailed sound descriptions for each sound-producing object. At a micro level, the MLLM needs to capture the duration of sound-producing objects in the video, segment them by time, and obtain detailed information on the movement trajectory of the sound-producing objects (e.g., whether they are on the left or right side of the screen, how far they are from the camera, etc.). Additionally, for specific tasks, the original video can be sliced for querying. In the audio generation phase, the obtained audio prompts are optimized and then input into a pre-trained audio generation model to generate the audio. Finally, the audio from the video understanding phase is processed and integrated to generate an audiovisually consistent video.

Our main contributions are as follows:

- **Methodology.** We propose V2A-CoT, to the best of our knowledge, the first training-free V2A method, which enhances video understanding from macro and micro perspectives through expert CoT, thus generating audio that is highly consistent with the video content.

- **Effectiveness and superiority.** Experimental evaluations show that, compared to state-of-the-art V2A methods, V2A-CoT enhances MLLM's audiovisual understanding ability, improving video understanding accuracy by up to 14% and nearly doubling the audiovisual consistency of the generated audio.

- **Scalability and Generality.** We demonstrate that V2A-CoT performs well even in finely controlled sound tasks through specific dubbing tasks. Furthermore, our method does not require training or fine-tuning of MLLMs and can be extended to various MLLMs.

## 2 RELATED WORK

### 2.1 CHAIN-OF-THOUGHT

In tasks involving understanding and reasoning, generating a "Chain of Thought" as intermediate reasoning steps significantly enhances the reasoning capabilities of LLMs. This concept was first introduced by Wei et al. (2022), who demonstrated how providing a few CoT examples in the prompt can guide language models to perform complex reasoning. Their work has provided crucial insights for subsequent research in the relevant area (Feng et al., 2024; Yao et al., 2024; Park et al., 2023). Mitra et al. (2024) extended the CoT approach by proposing compositional Chain-of-Thought (CCoT), a zero-shot prompting method that incorporates scene graph (SG) representations. CCoT generates an SG using the MLLM and uses it in the prompt to improve compositional visual reasoning. Their research shows that CCoT enhances MLLM's performance on both compositional and general multimodal benchmarks. Similarly, Cantor, proposed by Gao et al. (2024), extends the CoT framework by introducing a multimodal approach that combines perception and decision generation. Cantor utilizes the advanced cognitive capabilities of MLLMs to enhance visual reasoning, achieving significant improvements on complex datasets without requiring fine-tuning or ground-

truth rationales. There are also some efforts in the video field that try to use CoT to improve their work (Wang et al., 2024b; Fei et al., 2024).

## 2.2 VIDEO UNDERSTANDING TECHNIQUES

Early video understanding techniques were mostly based on traditional models (e.g., convolutional neural network), with additional methods developed for temporal processing (Huang et al., 2018; Lin et al., 2019; Bertasius et al., 2021). However, with the advancement of multimodal models, large models specifically designed for video understanding have gradually been developed (Lyu et al., 2023; Shu et al., 2023; Tang et al., 2025). Video-ChatGPT (Maaz et al., 2023) introduces a multimodal conversation model that combines a video-adapted visual encoder with an LLM, enabling detailed conversations about videos. Video-LLaVA (Lin et al., 2023) introduces a unified vision-language model that integrates visual representation into the language feature space, overcoming the challenges of misalignment between image and video tokens. By learning from a mixed dataset of images and videos, it mutually enhances both modalities. LLaVA-NeXT-Interleave (Li et al., 2024) extends the capabilities of MLLMs by simultaneously addressing multi-image, multi-frame (video), multi-view (3D), and multi-patch (single-image) tasks. The development of these large models for video understanding paves the way for subsequent advancements in video understanding and generation technologies.

## 2.3 VIDEO-TO-AUDIO

To generate more realistic videos, several works have developed V2A methods based on large models (Wang et al., 2024c; Haji-Ali et al., 2024). Diff-Foley (Luo et al., 2024) introduces a synchronized Video-to-Audio synthesis method using a latent diffusion model (LDM), enhancing audio generation quality through improved temporal synchronization and audio-visual relevance. TiVA (Wang et al., 2024a) presents a novel time-aligned V2A framework that jointly focuses on semantic matching and temporal synchronization in audio generation. By separately encoding visual semantics and predicting an audio layout, TiVA uses latent diffusion to produce audio more closely aligned with the video. Stable-V2A (Gramaccioni et al., 2024) presents a two-stage model designed to assist sound designers and Foley artists by automating repetitive tasks in audio production while preserving creative control. The model consists of an RMS-Mapper, which estimates an audio envelope aligned with the video, and Stable-Foley, a diffusion model that generates semantically and temporally accurate audio. MaskVAT (Pascual et al., 2025) introduces a V2A generative model that combines a high-quality audio codec with a sequence-to-sequence masked generative model, enabling simultaneous high-quality audio, semantic matching, and temporal synchronization. A key characteristic of such works is that they typically require model training, however, the associated costs are often prohibitive for many researchers and practitioners.

# 3 V2A-CoT

The V2A-CoT we propose is a complete process for video dubbing. First, we describe the standard process of video understanding using an MLLM. Then, we introduce V2A-CoT, which is divided into two parts: the first part involves video question-answering through a zero-shot expert CoT, specifically generating an object list, predicting sounds, and determining sound details; the second part is audio generation. Our overall framework is shown in Figure 2.

## 3.1 PRELIMINARIES: VIDEO QUESTION ANSWERING

Video question answering (Video-QA) is mainly conducted through MLLMs. Generally speaking, in order for the large model to understand the content without consuming too many resources, it is necessary to first sample a collection of frames of the video $S_f$, as shown in the following formula:

$$S_f = \{frame_i \mid i \in \mathcal{I}, frame_i \in \mathcal{V}\} \tag{1}$$

where $\mathcal{I}$ represents the number of frames in video $\mathcal{V}$, and $frame_i$ is the $i$-th frame in the video.

Then, the set of obtained frames $S_f$ and the problem $\mathcal{Q}$ are sent to MLLM in a certain format, and a response is obtained. This process can be represented by the following formula:

$$\mathcal{A} = \mathcal{F}_{MLLM}(S_f, \mathcal{Q}) \tag{2}$$

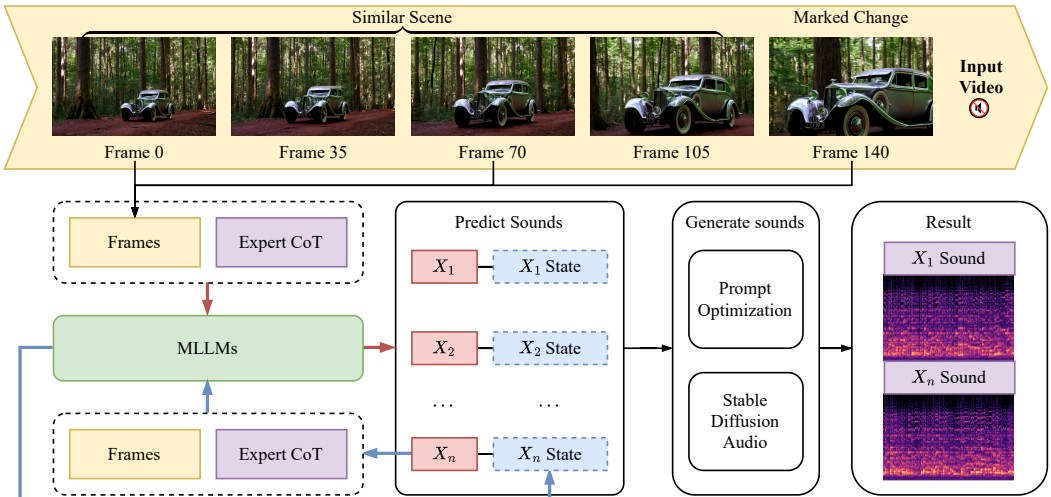

Figure 2: The overall framework of V2A-CoT. Firstly, it is necessary to perform frame processing on the video to grasp its content accurately. Secondly, expert CoT is used to obtain macro sound deconstruction. Then, the main sound elements obtained $X_i$ are judged for sound changes. Finally, prompt optimization and sound synthesis are carried out through pre-trained latent diffusion models.

where $\mathcal{F}_{MLLM}(\cdot)$ represents MLLM function, and $\mathcal{A}$ is the answer returned by the MLLM.

Although the number of frames and question answering quality supported by different MLLMs may vary, some of them still have targeted purposes (such as for motion capture, object tracking, etc.), but generally speaking, video question answering is conducted through the above methods. Multimodal reasoning tasks are different from general logical reasoning tasks. Current LLM reasoning mainly solves certain problems, such as mathematical reasoning, string symbol reasoning, etc. The model only needs to solve the problem itself. However, for multimodal problems, it is often difficult to have a definite answer due to the influence of the input (for example, even the most advanced multimodal video understanding model can only input a limited number of frames of a video), and the problem to be solved includes the problem and the input video, so it is more challenging.

## 3.2 KEY FRAMES EQUALIZATION

As mentioned earlier, existing MLLMs understand videos by inputting key frames from the video and comprehending the video content through these key frames. The extraction of key frames can be divided into two steps. First, a large threshold is used to trim the scenes with significant changes in the video frames, obtaining video segments with smaller scene changes (scenes with large changes are typically considered as edited videos or spliced clips from different video segments). This step is designed to allow the MLLM to capture as much key information as possible, while some repetitive information can be appropriately ignored to reduce inference costs. Second, the video segments with smaller scene changes after trimming are further processed. If the key frames extracted from the video span a large range, additional frames are added as necessary (for example, if the frames from the 1st to the 600th are similar, additional frames in this range will be added). This is to prevent the loss of details due to a large frame span. This method allows the MLLM to understand the video at a macro level without losing crucial details.

## 3.3 EXPERT CHAIN-OF-THOUGHT

Inspired by human synesthesia and the ability to describe sounds through language, the proposed V2A-CoT consists of two steps: *1) Macro sound deconstruction:* For a given video audio problem to be solved, the original question along with the prompt is input into an MLLM to obtain the answer. For judgment or quantity questions, accurate values should be provided, while for divergent questions, hallucination should be avoided, and more reasonable results should be included. *2) Micro sound judgment:* After obtaining the specific sound from step *1)*, the sound needs to be synthesized by judging the detailed variations in sound (e.g., changes in volume over time) to generate more

| $Q_1$ | *What is the first instrument that comes in?* | $A_1$ | *Piano* |

| **MLLM** | **MLLM + Standard CoT** | **MLLM + V2A-CoT (Ours)** |
|---|---|---|
| **Piano** | The video shows a man and woman playing the piano in a room. The first instrument to come in is the acoustic guitar. | The video shows a man and woman playing music in a room. **The first instrument to come in is the piano**, followed by the cello. The video also shows a skateboard in the background. |

| $Q_2$ | *Is the instrument on the right more rhythmic than the instrument on the left?* | $A_2$ | *No* |

| **MLLM** | **MLLM + Standard CoT** | **MLLM + V2A-CoT (Ours)** |
|---|---|---|
| Yes, there are piano and suona instruments in the video. They are both sounding at the same time as the trumpet. | The video shows a man in a suit playing the trumpet in front of a wall. **There are no piano or suona instruments in the video.** | The video shows a man in a suit playing the trumpet in front of a wall. **There are no piano or suona instruments in the video.** |

Figure 3: Comparison of the effectiveness of three different questioning methods (direct questioning, standard CoT, and V2A-CoT) during video comprehension, where the performance of standard CoT is not always ideal, while V2A-CoT proves to be more stable. $Q\&A$ refers to question-answer pairs.

realistic audio. As shown in Figure 3, a comparison of different questioning methods illustrates the clear advantage of V2A-CoT.

*Macro sound deconstruction.* In order to make the MLLM think like a human and fully understand the video content, V2A-CoT needs to guide the MLLM through the following thought process:

- **Part 1: Scene recognition.** In visual content analysis, the first step is to answer the question, "What kind of environment is this?" Different scenes usually correspond to distinct sound libraries. For instance, urban streets are typically associated with the sounds of cars and crowds, while natural environments like forests are characterized by wind and birdsong. Scene recognition helps narrow the scope of sound inferences, making subsequent predictions more focused and accurate.

- **Part 2: Objects.** The most direct sounds in a video often come from primary objects and their actions (such as talking, singing, or knocking). The MLLM needs to convert visual actions into sound sources and identify the sound-producing objects and their positions. For example, if a person is playing guitar on stage, it corresponds to the sound of a guitar.

- **Part 3: Environment.** Scene recognition in **Part 1** offers only a general estimate, lacking the ability to identify specific environmental sounds. In this step, V2A-CoT directs the MLLM to take into account the impact of environmental factors on sound. For example, the hum of air conditioning or household appliances, echoes in valleys, or the sound of ocean waves.

- **Part 4: Interaction.** To make the sounds added to the video more realistic, V2A-CoT guides the MLLM to think about the sounds produced by interactions between objects and between objects and the environment. For example, when a vehicle is driving on a dirt road, object reasoning in **Part 2** may infer that the car (engine) will make a sound, but interaction analysis can further infer that the friction between the tires and the ground will also produce a sound. Even in a static image, by capturing posture or action moments, potential physical sounds can be inferred (e.g., a person about to close a door, leading to the inference of the sound of the door closing).

- **Part 5: Common sense.** Relying solely on a limited number of visual frames may not reveal all details, but based on contextual knowledge, the MLLM can supplement or predict hidden or common sounds, enhancing the realism of the video. For example, when a video

depicts a bar stage with a shot of dancers, it can be inferred that there should be noise from the surrounding crowd, even if they are not shown in the video.

- **Part 6: Listing.** The final step is to guide the MLLM to provide a list of the most likely sounds that could occur in the current video, making subsequent processing and sound synthesis easier.

The overall input for obtaining macro level sound can be represented as follows, where $P_1$ to $P_6$ correspond to the six parts mentioned above, $Q$ represents the original question, $Q_{in}$ represents the final question, $I_a$ represents auxiliary information (e.g. "To answer this question, you should..."), and $S_f$ represents the set of extracted video frames:

$$Q_{in} = \text{"}[S_f][Q][I_a][P_1, P_2, \ldots, P_6][Q]\text{"} \tag{3}$$

Here, $Q$ appears twice to emphasize or reinforce memory, a technique similar to repeated questioning commonly seen in previous works (Mitra et al., 2024).

*Micro sound judgment.* Due to the fact that macro sound deconstruction can only list the sounds that may be present in the video, knowing these sounds alone is not enough to synthesize realistic audio. This is because some sounds in the video are often dynamically changing. As shown in Figure 4, we can further inquire about the relative position changes of the main objects and camera angles in the MLLM to infer the variations in sound.

### 3.4 OPTIMIZATION AND SYNTHESIS

Since our dubbing focuses on natural sounds, we define rules to optimize the prompts. For instance, we specifically annotate "avoid musical melodies" and "no human dialogue" (since

Figure 4: An example of micro sound judgment. The motorcycle in the video moves from far to near relative to the camera.

such content requires specific fine-tuning). Once the rules are defined, the LLM can automatically refine the descriptions of primary and environmental sounds extracted from video understanding. Finally, these optimized prompts are input into a pre-trained sound generation model (such as Stable Diffusion Audio v1) to generate the final audio, which is further fine-tuned based on the features needed to synthesize realistic video content for each sound.

By using V2A-CoT, we can generate high-quality video and audio samples. Specifically, we first employ existing video generation models to create a series of AI videos with different categories and prompts, then automatically generate sound descriptions matching the video content using V2A-CoT. These descriptions are then used to generate the complete audio, which is integrated into the video. The samples generated in this process include audio, video, sound descriptions, and Mel spectrograms of the sound. Leveraging V2A-CoT enables the seamless generation of high-quality video samples, which holds significant potential in the field of multimodal audiovisual learning.

## 4 EXPERIMENTAL EVALUATION

Since the V2A-CoT method is divided into two parts: video understanding and audio generation, we need to evaluate the effectiveness of these two parts separately. Through comparisons and ablation experiments, we demonstrate that V2A-CoT offers significant advantages in both video understanding and audio generation.

### 4.1 EXPERIMENTAL SETUP

**Testbed.** We evaluate V2A-CoT on a cloud server using an NVIDIA A40 GPU. The system runs Ubuntu 22.04, with PyTorch version 2.1.0 and CUDA version 12.1.

Table 1: The impact of the number of different input frames.

| Methods | Frame Number | | | | | | |
| --- | --- | --- | --- | --- | --- | --- | --- |
| | 2 | 4 | 8 | 12 | 16 | 32 | 64 |
| Video-LLaVA (Lin et al., 2023) | 57% | 57% | 60% | 69% | - | - | - |
| VideoLLaMA-2 (Cheng et al., 2024) | 60% | 69% | 76% | 80% | 85% | 88% | - |
| LLaVA-NeXT(Video) (Li et al., 2024) | 67% | 70% | 75% | 78% | 79% | 76% | 74% |

Table 2: The performance results of our method on different types of problems.

| Methods | Problem Types | | | | |
| --- | --- | --- | --- | --- | --- |
| | Counting | Existential | Location | Comparative | Temporal |
| Video-LLaVA (Lin et al., 2023) | 65% | 67% | 78% | 83% | 50% |
| VideoLLaMA-2 (Cheng et al., 2024) | 82% | 100% | 89% | 100% | 75% |
| LLaVA-NeXT(Video) (Li et al., 2024) | 82% | 67% | 89% | 67% | 50% |
| Average | 76% | 78% | 85% | 83% | 58% |

**Models.** The MLLMs we use include VideoLLaMA-2 (Cheng et al., 2024), the video understanding version of LLaVA-NeXT (Li et al., 2024), Video-LLaVA (Lin et al., 2023), and GPT-4o and GPT-4o-mini for video understanding by inputting image sets. These large models are currently state-of-the-art (SOTA) for video understanding and were not retrained or fine-tuned for the experiments.

**Baseline.** In the video understanding part, we compare V2A-CoT with two baselines. First, to evaluate the additional benefits of our method on pre-trained MLLMs, the baseline is applying the model to direct questioning without any other prompts. Second, we consider the zero-shot CoT baseline (Kojima et al., 2022) to determine the advantages of V2A-CoT compared to the standard CoT prompting method. In the audio generation part, we compare the audio generated by V2A-CoT with the audio generated by the training-based Stable-V2A (Gramaccioni et al., 2024) to assess the superiority of V2A-CoT.

**Dataset.** The primary dataset used in this evaluation is Music-AVQA (Li et al., 2022), a large-scale music performance dataset. All question-answer pairs in the dataset are divided into 3 modal scenes, including 9 question types and 33 question templates. The videos are all 60 frames per second, with a total duration of 60 seconds. In this dataset, the videos contain sound, and most of the questions require answering by combining both video and audio. Therefore, answering questions using only video understanding is very challenging, and the MLLMs we used only have the capability to understand video, not audio. It is important to note here that, although this dataset may be related to musical melodies, we only use it to judge and understand sound rather than generate sound, which does not contradict our previous optimization measures. Additionally, we also used the Greatest Hits dataset (Owens et al., 2016), which consists of videos where a wooden stick strikes various materials, presenting a significant challenge for MLLM's audiovisual understanding.

**Metrics.** We primarily use two metrics for evaluation. The first is accuracy, which represents the video understanding capability of the MLLM in the video understanding phase. The second is the Frechet Audio Distance (FAD), a reference-free evaluation metric for audio (Kilgour et al., 2018), which is also commonly used in previous works (Wang et al., 2024c; Haji-Ali et al., 2024). Compared to traditional audio metrics, FAD focuses on the perceptual quality of sound to humans, and using it as a metric allows for a quantitative analysis of the quality of the generated audio.

## 4.2 OVERALL PERFORMANCE

In terms of video understanding, we evaluate V2A-CoT on five models. As shown in Figure 5, this is a comparison of the accuracy of three questioning methods: direct questioning, standard CoT, and V2A-CoT. In this case, the prompt for the standard CoT method is constructed by appending the reasoning trigger "Let us think step by step." to the question, which generates linguistic reasoning for answering the question. Compared to direct questioning and standard CoT, V2A-CoT achieves better question-answer accuracy on multiple MLLMs. Specifically, an accuracy increase of about

14% is achieved on GPT-4o and GPT-4o-mini. This suggests that specialized CoT methods are more targeted in vertical domains. We also find that the advantages of V2A-CoT in video understanding are constrained by the performance of the MLLM itself. Furthermore, we observe that without using V2A-CoT, GPT-4o has a lower understanding accuracy than GPT-4o-mini,

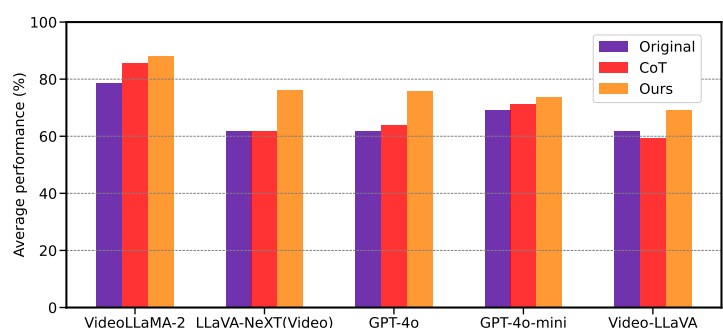

Figure 5: Comparative experiment. The accuracy results of V2A-CoT compared with the other two methods.

which may be due to the more severe hallucination phenomenon in GPT-4o, limiting its understanding capability. However, when using V2A-CoT, the accuracy of GPT-4o's understanding surpassed that of GPT-4o-mini, indicating that V2A-CoT mitigates hallucinations to some extent.

### 4.3 ABLATION STUDY

We conduct ablation experiments by adjusting the number of input video frames and question types to further evaluate the effectiveness of V2A-CoT. The results in Table 1 show the performance of V2A-CoT under different frame inputs, with missing parts representing the input limitations of the corresponding MLLM. Theoretically, more input frames mean more information, which enables the MLLM to capture more details and understand the video content better, which is fundamentally different from image understanding. However, the results in Table 1 indicate that increasing the number of video frames does not always lead to better results. This might be because more frames increase the inference time, which could amplify distracting information and potentially intensify hallucinations. Each model has an optimal number of input frames. Table 2 shows the performance of our method across different question types. It can be observed that current MLLMs have fairly average capabilities in time judgment, which also highlights the significance of temporal understanding guidance, such as object interaction and dynamic analysis, in the V2A-CoT framework.

### 4.4 AUDIO QUALITY AND AUDIOVISUAL CONSISTENCY

To evaluate the audio quality generated by the V2A-CoT framework compared to training-based methods, we compared the audio generated by both approaches. Specifically, we used V2A-CoT and Stable-V2A to generate audio for the same silent video to compare their audiovisual consistency. Our method achieved an FAD score of 8.98, while Stable-V2A scored 16.57 (a lower value indicates higher similarity between the generated audio and the original reference audio, resulting in better audiovisual consistency). The audiovisual consistency metric nearly doubles, demonstrating the superiority of V2A-CoT. Additionally, since training-based methods typically require more computational and time resources, V2A-CoT also offers a cost-effective advantage.

### 4.5 SOUND-TIME MATCHING

Since V2A-CoT is a method for extracting sound from silent videos, we introduce a special dubbing task to validate that our method can effectively handle challenging tasks. The dataset for this task is the Greatest Hits (Owens et al., 2016), where the video content shows a small wooden stick striking objects made from various materials. These videos are categorized based on the material type of the object being struck, such as fabric, grass, and water. These videos contain the original audio of the stick

Table 3: Accuracy of sound judgment when hitting different types of objects with a wooden stick.

| Category | Accuracy (%) | Seconds (s) |
|---|---|---|
| Cloth | 100% | 5 |
| Grass | 88.89% | 5 |
| Gravel | 93.75% | 5 |
| Plastic bag | 77.50% | 5 |
| Water | 70.24% | 5 |
| Wood | 77.38% | 5 |
| Average | 84.63% | 5 |

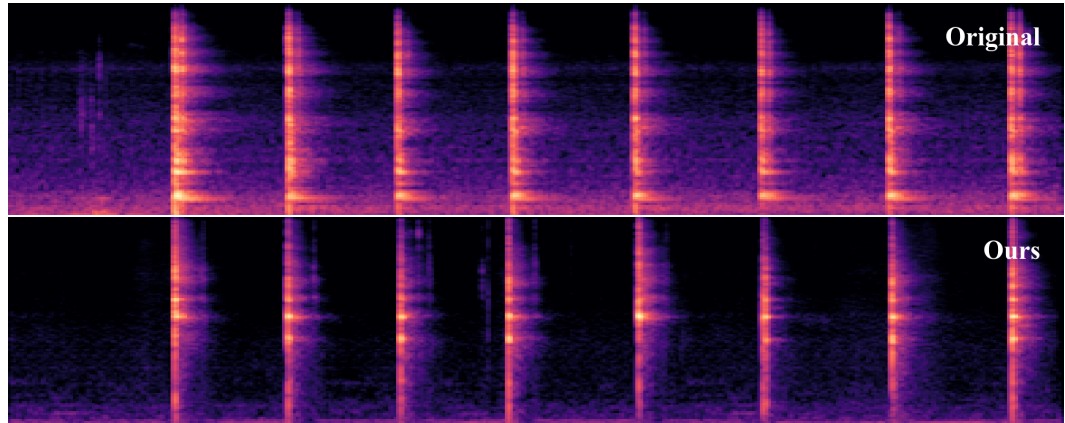

Figure 6: Visual comparison of the sound of small wooden sticks striking. The upper part is the original video sound, and the lower part is the synthesized sound.

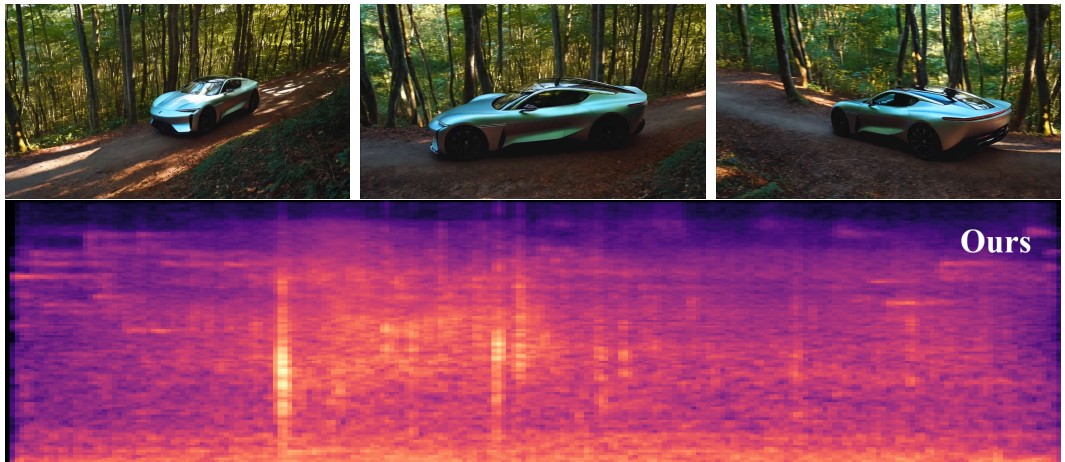

Figure 7: A visualization example of AI videos we generated. Note that we synthesize the sound inside, and the original generated AI video has no sound.

hitting the objects. Therefore, we use V2A-CoT to dub silent versions of the videos and compare them to the original audio to evaluate the effectiveness of V2A-CoT. The results in Table 3 and Figure 6 show that, both quantitatively in terms of accuracy and qualitatively through the Mel spectrograms, V2A-CoT demonstrates high accuracy and is capable of producing sounds corresponding to the materials at the correct time points. Finally, we demonstrate the use of V2A-CoT to generate AI videos with sound. As shown in Figure 7, we present an example of the generated video.

## 5 CONCLUSION

In this paper, we propose V2A-CoT, a framework for dubbing silent videos, which, to the best of our knowledge, is the first training-free V2A framework. V2A-CoT consists of two main components: first, it simultaneously understands video content from both macro and micro perspectives, focusing on sound sources and their dynamic changes; second, it generates highly consistent audiovisual audio by capturing this information. We validate the effectiveness and superiority of V2A-CoT through various experiments in video understanding and audio generation. Whether for general video understanding or specific dubbing tasks, V2A-CoT outperforms existing methods. While training-based methods require extensive training, introducing cost issues, our method's significant advantage is that it enables dubbing without the need for training.

## ETHICS STATEMENT

We recognize the potential for biases in the development and application of multimodal AI technologies, particularly in the creation of video and audio content. The design and implementation of V2A-CoT may inadvertently reflect certain assumptions or perspectives that are not universally applicable or may exclude diverse cultural or societal contexts. We are also aware of the potential for misuse, particularly in the creation of deceptive or harmful media. While our goal is to enhance accessibility and creativity in content generation, we acknowledge that malicious actors may exploit these technologies to produce misleading or damaging media, contributing to the spread of misinformation. Therefore, we are committed to ensuring the responsible use of our technology, taking active steps to prevent misuse and promoting ethical deployment.

## REPRODUCIBILITY STATEMENT

We provide a comprehensive description of the proposed framework, including the principles, effects, and technical details of each component. The datasets used in this work are all open-source. Portions of the code, experimental setups, and examples are available in the supplementary materials to facilitate reproducibility.

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

## A  PROMPT EXAMPLES

These are two prompt examples of V2A-CoT, which need to be used together with the original question to generate specific information of the sound.

> To answer the question, please think as follows:
>
> 1. Please describe the overall scene presented in the video/images. Based on your understanding of the scene, list the common sounds that may be related to the scene from a common-sense perspective.
>
> 2. What are the main characters or objects in the picture? What actions are these subjects taking? Based on their actions, what sounds might be produced?
>
> 3. Observe the background environment in the video/images, and combine environmental and background clues to infer the possible ambient sounds in the scene.
>
> 4. If people or objects are colliding or interacting in the picture, please list the possible sources of sound.
>
> 5. Based on your understanding of the social culture or life experience of the scene, what common sounds usually appear in such scenes?
>
> 6. Integrate the above analysis. First, list the main possible sounds, and then add secondary or less certain sounds. Based on the current picture, give a list of the most likely sounds and explain the confidence or priority if necessary.

> To answer the question, please think as follows:
>
> 1. List possible background sounds based on the scene and environment in the picture.
>
> 2. Analyze the characters or objects in the picture, their movements, and the main sounds that may be emitted.
>
> 3. What obvious interactions or collision actions are noticed, and what kind of sounds may be produced?
>
> 4. Combining common sense and cultural background, add some less obvious but common voices.
>
> 5. If there are any uncertain details, please list different possibilities separately and indicate the reasons.
>
> 6. Finally, please provide an integrated list of sounds, including primary and secondary sounds, and provide explanations for areas where you feel uncertain.

## B  USAGE OF LLMS

In this work, LLMs are only used as components of the process. During the writing of the paper, we use LLMs to polish the text. Apart from this, any innovative designs and ideas are original and not derived from LLMs.

## C  LIMITATIONS

This work is not without limitations, primarily due to the inherent constraints of the MLLM itself. First, when dealing with low-resolution videos, the MLLM may face challenges in accurately identifying the names or states of objects in the video. For example, similar objects may be confused, leading to a slight decrease in recognition accuracy. Second, some MLLMs are limited in the number of video frames they can process. Fewer input frames may result in an incomplete understanding

of the video, leading to a decline in accuracy. These challenges could potentially be mitigated by using more powerful MLLMs.

## D    BROADER IMPACTS

This work aims to promote the development and application of multimodal AI technologies, particularly in the field of video and audio generation. V2A-CoT conveniently improves the quality of audiovisual content creation, which has the potential to democratize AI media production, allowing creators, educators, and professionals from various domains to generate high-quality content without the need for expensive resources or specialized training. This can foster innovation and reduce barriers for individuals and organizations in using traditional media production tools.

However, the widespread use of such powerful generative technologies also raises significant ethical concerns. The ability to create realistic, high-fidelity audiovisual content with minimal effort could be misused to produce deceptive or harmful media, leading to the increase of misinformation, deepfakes, and other forms of synthetic media, thereby undermining public trust. Therefore, while V2A-CoT offers significant benefits in terms of accessibility and creativity, it is crucial to implement safeguards to prevent misuse and ensure the ethical deployment of this technology.

