# OpenReview forum: "V2A-CoT: A Training-Free Video-to-Audio Method via Expert Chain-of-Thought"
_ICLR.cc/2026/Conference — Submitted to ICLR 2026_

### Official Review · Reviewer_TfVY · 2025-10-16

**Soundness:** 3
**Presentation:** 3
**Contribution:** 2
**Rating:** 2
**Confidence:** 4

**Summary:**

Paper introduces a training-free video-to-audio (V2A) framework that enhances the realism of silent videos by generating synchronized audio. ​ Unlike existing V2A methods that require extensive training, V2A-CoT leverages multimodal large language models (MLLMs) and a zero-shot Chain-of-Thought (CoT) approach to understand video content from macro and micro perspectives. ​ The framework consists of two main components: video understanding and audio generation. ​ It uses expert CoT to analyze scenes, objects, environments, interactions, and common sense to predict sound attributes. ​ The audio is then synthesized using pre-trained models, ensuring high audiovisual consistency. ​ Experimental evaluations demonstrate that V2A-CoT improves video understanding accuracy by up to 14% and nearly doubles audiovisual consistency compared to state-of-the-art methods. ​ V2A-CoT is scalable, cost-effective, and adaptable to various MLLMs without requiring additional training, making it a significant advancement in multimodal AI technologies for video dubbing and audiovisual content creation. ​

**Strengths:**

1. V2A-CoT does not require extensive training. ​

2. V2A-CoT improves video understanding accuracy by up to 14% compared to state-of-the-art methods, leveraging a zero-shot Chain-of-Thought (CoT) approach to analyze video content from both macro and micro perspectives. ​

3. V2A-CoT nearly doubles the audiovisual consistency of generated audio compared to training-based methods, as demonstrated by Frechet Audio Distance (FAD) evaluations. ​

4. The method is plug-and-play, seamlessly integrating with various multimodal large language models (MLLMs) without requiring fine-tuning or retraining. ​

**Weaknesses:**

1. Only relying on FAD may not be sufficient. Other common metrics include Inception Score (IS), Frechet Distance (FID) and Mean KL Divergence (MKL) should be considered. No temporal accuracy metric is used to evaluate the generated audio.

2. Insufficient comparison with other Video-to-Audio baselines. Although the work focuses on audio generation, most evaluations focus on video understanding.

3. Performance is constrained by the inherent limitations of the multimodal large language models (MLLMs) it relies on, such as their ability to process low-resolution videos or accurately identify objects and their states. ​

4. Some MLLMs have restrictions on the number of video frames they can process, which may lead to incomplete video understanding and reduced accuracy when fewer frames are analyzed. ​

5. The framework may struggle with accurately identifying objects or their states in low-resolution videos, leading to potential errors in sound generation. ​

**Questions:**

1. In section 3.4, what are the detailed rules to optimize the prompts?

2. How to ensure the consistency between each sound segment (X_1 to X_n in Figure 2)? The author mentioned Micro sound judgment to changes in volume over time. However, there could be potential issues with consistency since the audio is not generated in a single pass inference. Combining segments into one audio file may lead to inconsistency problems.

3. How does V2A-CoT handle scenarios where the video contains ambiguous or unclear visual information, such as low-resolution or heavily occluded objects?

4. What are the specific challenges faced when integrating V2A-CoT with different MLLMs, and how does the framework address these challenges? Any ablations on different MLLMs?

5. How does V2A-CoT ensure the accuracy of sound generation for complex or less common scenes (e.g. out-of-domain) where environmental sounds are not easily inferred?

6. How does V2A-CoT mitigate the hallucination phenomenon observed in certain MLLMs, and are there specific techniques used to improve the reliability of the generated sound descriptions? ​

7. How does V2A-CoT perform on videos with longer durations or higher frame rates, and are there plans to optimize the framework for such cases?

**Details Of Ethics Concerns:**

No notable ethics concerns

---

### Official Review · Reviewer_RThz · 2025-10-29

**Soundness:** 2
**Presentation:** 1
**Contribution:** 1
**Rating:** 2
**Confidence:** 4

**Summary:**

This paper proposes V2A-CoT, a method for generating audio for silent videos in a "training-free" manner. The core idea is to use a multimodal large language model (MLLM) guided by a detailed, multi-part "Expert Chain-of-Thought" (CoT) prompt. This prompt guides the MLLM to deconstruct the video from "macro" (scene, objects, interactions) and "micro" (relative positions) perspectives to generate a textual description of the expected sounds. This text description is then optimized and fed into a pre-trained text-to-audio model (like Stable Diffusion Audio) to synthesize the final audio. The experiments demonstrate the effectiveness of the proposed method.

**Strengths:**

- Exploring training-free methods for video dubbing is an interesting idea.
- The source code is provided in the supplementary material.

**Weaknesses:**

The quality of this manuscript is insufficient for acceptance.
- The novelty of this paper is limited. The idea of leveraging CoT to obtain higher-quality prompts for a subsequent audio generation model is trivial. Furthermore, the generalizability and future prospects of this method are also concerning.
- The manuscript suffers from poor writing quality. The description of the proposed method is ambiguous and missing critical details, making it difficult to understand. Including a concrete illustrative example would likely improve the clarity of the methodology.
- The experimental setup lacks sufficient breadth. The authors should benchmark V2A-CoT on additional datasets—like VideoMME, AVUT, WorldSense, and DailyOmni—to provide stronger evidence for its utility in QA tasks. Moreover, the evaluation of the method should not be limited to the current models; stronger baselines or recent models like Qwen3-VL should also be tested.
- The validation of audio quality is too narrow, which is a significant weakness as this is the purported focus of the paper. The datasets and metrics employed are insufficient, and the subjective human evaluation is omitted.

**Questions:**

- "Specifically, an accuracy increase of about 14% is achieved on GPT-4o and GPT-4o-mini." cannot be tell in Figure 5.
- The description provided for Table 3 is insufficient.

---

### Official Review · Reviewer_hEqL · 2025-10-31

**Soundness:** 1
**Presentation:** 2
**Contribution:** 1
**Rating:** 2
**Confidence:** 5

**Summary:**

This paper presents V2A-CoT, a method for generating audio from silent video. The approach uses a Multimodal Large Language Model (MLLM) with a structured prompt, which the authors call an "Expert Chain-of-Thought," to produce a text description of the sounds that should be present in the video. This generated text is subsequently used as a prompt for a standard text-to-audio synthesis model to create the final audio. The authors position this as a "training-free" framework and claim it improves video understanding and achieves superior audio-visual consistency compared to a selected training-based baseline.

**Strengths:**

The paper investigates the direction of leveraging large pre-trained models to circumvent the need for expensive, task-specific training. Exploring such low-resource paradigms is a potentially interesting research avenue for the community.

**Weaknesses:**

Despite the simplicity of its premise, this paper is beset by major, disqualifying flaws, ranging from a critical lack of novelty to a deeply inadequate and misleading experimental evaluation.

1.  **Severe Lack of Novelty:** The core idea of this paper—using an MLLM to reason about video content and generate a textual prompt for a subsequent text-to-audio (T2A) model—is not new. This exact pipeline has been proposed and explored in recent, highly relevant works that are conspicuously absent from this paper's literature review and comparison. Specifically:
    *   **`SonicVisionLM`[1]** presents a nearly identical ideology of using a Vision Language Model to "play sound" by generating textual audio representations.
    *   **`ThinkSound`[2]** explicitly uses a Chain-of-Thought (CoT) reasoning process within an MLLM to generate and edit audio.
    The failure to cite, discuss, or differentiate from these works suggests a lack of thoroughness in the literature survey. As it stands, V2A-CoT appears to be a reimplementation or, at best, a marginal variation of already existing ideas, making its claimed contribution negligible.

2.  **Fundamentally Flawed and Inadequate Evaluation:** The experimental section is not rigorous enough to support any of the paper's claims and omits standard evaluation practices for this task.
    *   **Omission of SOTA Baselines:** The paper fails to compare against the actual state-of-the-art, namely `ThinkSound`. Comparing only to `Stable-V2A` (a training-based diffusion model) is an inappropriate and potentially misleading choice. Any claims of performance superiority are unsubstantiated without a direct comparison to its closest conceptual competitors.
    *   **Absence of Subjective Evaluation and Demos:** Audio generation is a perceptual task. Relying solely on an objective metric like FAD is insufficient. The lack of any human subjective evaluation (e.g., Mean Opinion Score for audio quality and audio-visual synchrony) is a critical omission. Furthermore, for a paper on audio generation, the absence of a demo page with audio-visual examples is unacceptable. It prevents reviewers and readers from verifying the claimed quality of the generated sounds, rendering the results unverifiable.
    *   **Questionable FAD Comparison:** The claim that a zero-shot prompting method nearly doubles the performance of a specialized, trained V2A model on the FAD metric is extraordinary and lacks evidence. The paper provides no details on the dataset, sample size, or specific configurations used for this comparison, making the result scientifically unsound and impossible to scrutinize.

3.  **Overclaimed Contribution and Lack of Rigorous Validation:** The "Expert Chain-of-Thought" is presented as a key contribution. However, it is essentially a manually-crafted, multi-step prompt template based on common sense (analyze scene, objects, interactions, etc.). While structured prompting is a useful technique, framing this as a novel reasoning framework is an overstatement. More importantly, **the authors make broad claims about their method's effectiveness ("excels in finely controlled sound tasks", "highly consistent audiovisual audio") but fail to provide the rigorous validation necessary to support them.** The evaluation is confined to simple, constrained datasets (`Greatest Hits`, `Music-AVQA`). There is no analysis demonstrating how the method performs on complex, real-world "in-the-wild" videos with multiple overlapping sound events, rapid scene changes, or subtle auditory cues. The authors should have validated their exaggerated claims with experiments on more challenging benchmarks or through extensive qualitative analysis of difficult cases to truly probe the limits and effectiveness of their approach. Without such evidence, the claimed capabilities remain unverified assertions.
4.  **Inherent Technical Limitations:** The method's reliance on a few keyframes and simple positional analysis is a naive approach to a complex problem. It cannot adequately model the continuous, dynamic, and overlapping nature of sounds in a real-world video. The temporal evolution of audio events (attack, decay, sustain, release) is lost, a domain where end-to-end trained models inherently have a strong advantage. The framework is merely a brittle pipeline of existing models, with its quality entirely capped by the limitations of the chosen MLLM and T2A model.

[1] Xie, Zhifeng, et al. "Sonicvisionlm: Playing sound with vision language models." Proceedings of the IEEE/CVF Conference on Computer Vision and Pattern Recognition. 2024.

[2] Liu H, Wang J, Luo K, et al. ThinkSound: Chain-of-Thought Reasoning in Multimodal Large Language Models for Audio Generation and Editing[J]. arXiv preprint arXiv:2506.21448, 2025.

**Questions:**

1.  The core conceptual pipeline of V2A-CoT appears nearly identical to that of `SonicVisionLM` and conceptually very similar to the CoT approach in `ThinkSound`. Can you precisely articulate the fundamental novelty of your work that justifies its publication in light of this significant prior art?
2.  `ThinkSound` is the most direct state-of-the-art competitor for a CoT-based V2A method. Why was it completely ignored in your evaluation? Without this crucial comparison, your performance claims are unsubstantiated.
3.  Why should the community accept the remarkable FAD score improvement over `Stable-V2A` as valid, given that the evaluation was conducted on an unspecified dataset, without a demo page for auditory verification, and lacks any form of subjective human rating, which is standard for this field?
4.  Given your claims of high performance and fine-grained control, can you provide results or analysis on challenging real-world videos that contain multiple, overlapping, and dynamic sound sources, which would be a more convincing test of your method's capabilities than the datasets currently used?

---

### Official Review · Reviewer_37nB · 2025-11-01

**Soundness:** 1
**Presentation:** 2
**Contribution:** 1
**Rating:** 2
**Confidence:** 4

**Summary:**

This paper proposes V2A-CoT—the first training-free Video-to-Audio (V2A) method based on MLLMs and expert CoT. After processing the video via key frame equalization, it understands the video from two perspectives: the macro perspective (encompassing 6 steps such as scene recognition, object/environment/interaction analysis) and the micro perspective (relative position of sound sources, dynamic changes). Subsequently, audio is generated through prompt optimization and a pre-trained model.

**Strengths:**

- This paper is well-motivated and easy to follow.
- It does not require fine-tuning of MLLMs and can be seamlessly adapted to various mainstream video understanding MLLMs such as VideoLLaMA-2 and GPT-4o.

**Weaknesses:**

- The paper generally lacks substantial technical novelty. It fails to propose breakthrough innovations in core links such as video understanding, audio generation, or the application of Chain-of-Thought (CoT). Most technical designs rely on existing frameworks (e.g., MLLM-based visual reasoning) or common-sense logical splitting, without forming unique technical contributions that distinguish it from current research.
- The authors deliberately weaken the baseline configurations in the experiment, which leads to an unfair comparison between the proposed method and baselines. This unfair experimental design cannot truly reflect the actual superiority of the proposed method, and the credibility of the experimental conclusions is thus greatly reduced.
- The "6-step macro sound deconstruction" (scene recognition → object analysis → environmental sound effect, etc.) in the paper essentially splits human common sense of audio-visual correlation into sequential steps, without introducing any new reasoning mechanisms or cross-modal alignment technologies. For micro sound judgment, only "capturing the relative position of sound sources" is mentioned, and no quantitative methods are proposed (e.g., how to convert the qualitative description of "closer to the camera" from MLLMs into specific audio volume parameters). Essentially, this is a simple application of the existing CoT framework, with no technical breakthroughs.
- The method for key frame selection in the paper is quite naive, with no innovative design in extraction strategies (e.g., no optimization for dynamic video content or complex scenes). More importantly, the paper fails to verify the effectiveness of the selected key frames—for example, whether these key frames can truly represent the core content of the video or contribute to improving the accuracy of subsequent video understanding—making the rationality of this step questionable.
- The number of MLLM baselines for video understanding compared in the paper is insufficient, and the selected baselines (e.g., Video-LLaMA2, Video-LLaVA) are all works from 2023 to 2024. The lack of comparison with more diverse or newer video MLLMs makes it impossible to fully evaluate the performance of the proposed method in the current state-of-the-art video understanding landscape.

**Questions:**

- Please explain why the answers of MLLM + standard CoT and MLLM + V2A-CoT are the same in certain test cases in Figure 3.
- For video understanding tasks where only video content is input (without any audio information), how does the method infer audio-related attributes? Taking the example in Figure 3—"Is the instrument on the right more rhythmic than the instrument on the left?"—rhythm is an inherent audio attribute. The paper needs to elaborate on the logical basis and specific reasoning process for inferring audio attributes like rhythm solely through visual information.
- In V2A-CoT, introducing additional key frames for video understanding does not effectively improve understanding performance. Essentially, this approach only incorporates the simpler task of image understanding into video understanding, without addressing the insufficient fine-grained understanding ability of MLLMs. Meanwhile, the overall performance of the proposed method is restricted by the image understanding ability of the underlying MLLMs. Please respond to this limitation and its impact on the method.
- What is the performance of the method in general video understanding tasks? The paper mainly focuses on video understanding related to audio generation, but lacks evaluation of the method’s performance in more general video understanding tasks (e.g., action recognition, scene classification, object tracking). Please supplement relevant experimental results or explain the applicability of the method in general video understanding scenarios.
- In Section 4.2 (Overall Performance), the paper only mentions evaluating "the video understanding capability of the MLLM in the video understanding phase" but does not specify key details of the benchmark, such as its name, dataset composition, and specific evaluation metrics (beyond accuracy). Please supplement these details to ensure the reproducibility and credibility of the experimental results.
- Using only the simple CoT prompt "Let us think step by step." deliberately weakens the baseline, leading to unfair comparison. Please address this issue.
- The V2A generation part of the paper adopts a simple video-text-audio pipeline and relies entirely on an external TTA model. This design is overly naive: it does not include a dedicated cross-modal alignment mechanism between video and audio, and the quality of generated audio is fully dependent on the performance of the external TTA model.
- How to ensure temporal alignment in V2A generation? The paper claims to generate audio corresponding to the video, but does not elaborate on the specific technical means to ensure temporal alignment between the generated audio and the video.
- The current video MLLM baselines (e.g., Video-LLaMA2, Video-LLaVA) are works from 2023 to 2024. There may be newer video MLLMs (published in 2025 or later) with more advanced performance. Comparing the proposed method with these latest video MLLMs will help better evaluate its competitiveness in the current research context. Please supplement such comparative experiments or explain the reasons for not doing so.
- Why are some entries blank in the frame number ablation experiment in Table 1?

---

### Meta-Review · Area_Chair_yokQ · 2026-01-04

**Summary:**

This paper is recommended for rejection due to a significant lack of novelty, as it closely resembles existing works like SonicVisionLM. The evaluation is flawed, omitting SOTA baselines, subjective human metrics, and audio demos, rendering the claimed performance gains unverifiable and the technical contribution negligible.

**Reviewer Concerns:**

None because the authors have not provided their responses towards the reviews.

**Reviewer Scores:**

The reviewers would like to maintain their score as the authors have not provided their responses towards the reviews.

---

### Decision · Program_Chairs · 2026-01-26

Reject